# Outcome and factors associated with undernutrition among children with congenital heart disease

**Indah K. Murni**[1,2]*, **Linda Patmasari**[1], **M. Taufik Wirawan**[1], **Nadya Arafuri**[1], **Neti Nurani**[1], **Esta Rossa Sativa**[1], **Sasmito Nugroho**[1], **Noormanto**[1]

**1** Department of Child Health, Dr. Sardjito Hospital/Faculty of Medicine, Public Health and Nursing, Universitas Gadjah Mada, Yogyakarta, Indonesia, **2** Center for Child Health-Pediatric Research Office (CCH-PRO), Faculty of Medicine, Public Health and Nursing, Universitas Gadjah Mada, Yogyakarta, Indonesia

* indah.kartika.m@ugm.ac.id

**Data Availability Statement:** All relevant data are within the paper and its Supporting Information file.

## Abstract

### Background

Congenital heart disease (CHD) is associated with high morbidity and mortality, especially among those with undernutrition. Evaluating risk factors associated with undernutrition is important to improve clinical outcomes. We aimed to evaluate the outcome and factors associated with undernutrition among children with CHD.

### Material and methods

A prospective cohort study was conducted among children with CHD at Dr. Sardjito Hospital, Yogyakarta, Indonesia during February 2016 to June 2018. Clinical and demographic data were collected at the time of diagnosis. Multivariate logistic regression analysis was used to identify independent factors associated with undernutrition using odds ratio (OR) and 95% confidence interval (CI). Significance was set as $p<0.05$.

### Results

We recruited 1,149 children with CHD, of those, 563 (49%) were underweight, 549 (47.8%) were stunting, and 361 (31.4%) were wasting. In the multivariate analysis, cyanotic CHD, delayed diagnosis, congestive heart failure, pulmonary hypertension, syndrome, young maternal age, history of low birth weight, and being first child were independently associated with undernutrition. Underweight and stunting were significantly associated with increased mortality with OR of 3.54 (95% CI: 1.62–7.74), $p<0.001$ and OR 3.31 (95% CI: 1.65–6.64), $p<0.001$, respectively.

### Conclusions

About half of the children with CHD were categorized with undernutrition. An increased risk of undernutrition was associated with cyanotic CHD, delayed diagnosis, congestive heart

**Funding:** The authors received no specific funding for this work.

**Competing interests:** The authors have declared that no competing interests exist.

failure, pulmonary hypertension, syndrome, low birth weight, and being first child. Underweight and stunting were significantly associated with increased risk of death.

## Introduction

Congenital heart disease (CHD) is the most common congenital anomaly in children [1]. Children with CHD tend to have an increased risk of undernutrition. The mechanism of undernutrition in children with CHD is compromised growth due to decreased energy intake and increased energy expenditure [2, 3]. In these children, undernutrition might impact in persistent impairment of somatic growth, increased susceptibility to certain infections including pneumonia, frequent hospitalization, poor cardiac surgery outcomes, developmental delay, poor school performance and reduced cognitive achievement, significant functional impairment in adult life and reduced work capacity, thus affecting economic productivity and increased death [2–4].

In Indonesia, undernutrition is one of the major problems in children and has contributed to increased mortality [5]. Assessment of nutritional status is important to measure the correct intake and feeding needed in order to provide the optimal growth of the children [6]. Few studies investigate the outcomes and predictors of undernutrition in children with CHD [7–9]. One study in a low and middle-income country revealed that undernutrition including wasting and stunting were prevalent and these were associated with severe heart failure and anemia in children with CHD leading to increased risk of death [7]. As far as we are aware, there are limited studies of the outcomes and factors associated with undernutrition among children with CHD in Indonesia. Accordingly, this study aimed to determine the incidence, outcomes and factors associated with undernutrition (underweight, wasting, and stunting) among children with CHD in Yogyakarta, Indonesia.

## Materials and methods

A prospective cohort study was conducted at Dr. Sardjito Hospital in Yogyakarta, Indonesia during February 2016 to June 2018. Children with CHD attending the pediatric cardiology outpatient clinic, pediatric wards, Pediatric Intensive Care Unit (PICU), perinatology ward, and Neonatal Intensive Care Unit (NICU) were included in the study.

Clinical and demographic data included age, sex, being a first child, low maternal education, the presence of syndrome, low birth weight, cardiac lesion type, congestive heart failure (CHF), pulmonary hypertension (PH), and delayed diagnosis of CHD. Low birth weight was defined as birth weight under 2,500 grams. Low maternal education was considered as high school or below. Presence of genetic syndrome was defined when patients were diagnosed as having any congenital dysmorphic syndrome. The diagnosis of syndrome was defined based on clinical appearance of the patients.

Delayed diagnosis was defined based on their pathologic type: cyanotic CHD and acyanotic CHD. Delayed diagnosis in acyanotic CHD was defined at an age where elective cardiac repair should have already been performed or in case the immediate treatment was indicated because of the patient hemodynamic status. Patients with delayed diagnosis in cyanotic CHD were defined as newborns discharged from their birth clinic or hospital without a CHD diagnosis [10].

The nutritional status was determined using criteria which were derived from the standards of the World Health Organization (WHO). Undernutrition included underweight, stunting and wasting. Categorical data were represented as frequencies and percentages. Univariate analysis was performed to determine the significance and strength of the association between

each factor and undernutrition as well as undernutrition and mortality. We assessed significance by Chi-square tests for categorical variables and $p<0.05$ was considered to indicate statistical significance. Multivariate analysis was conducted to determine factors that were independently associated with undernutrition. We selected all potential factors, including all variables found to be significant in the univariate analysis, and entered them into a multivariate logistic regression analysis. The results of multivariate analysis were reported as adjusted odds ratios (OR) with 95% confidence intervals (CI). Data were analyzed using STATA version 12.1 (StataCorp LP, College Station, Texas, USA).

The Medical and Health Research Ethics Committee of Universitas Gadjah Mada, Indonesia approved this study (KE/FK/0607/05/2019). Written informed consent was obtained from parents/guardians included in the study.

The primary outcome of this study was the outcome of undernutrition among children with CHD. The secondary outcome were factors associated with undernutrition among children with CHD.

## Results

There were 1,149 children with CHD included during the study period, and 530 (46.1%) were male (Table 1). Of those, 563 (49%) were underweight, 549 (47.8%) were stunting, and 361 (31.4%) were wasting (Fig 1).

**Table 1. Baseline characteristics among children with congenital heart disease.**

| Characteristics | n = 1149 (%) |
|---|---|
| Age in years | |
| 0–5 | 889 (77.4) |
| 5–10 | 137 (11.9) |
| >10 | 123 (10.7) |
| Sex | |
| Male | 530 (46.1) |
| Female | 619 (53.9) |
| Types of congenital heart disease | |
| Acyanotic | 936 (81.4) |
| Cyanotic | 213 (18.5) |
| Acyanotic congenital heart disease | |
| Ventricle septal defect | 312 (27.2) |
| Atrial septal defect | 233 (20.3) |
| Persistent ductus arteriosus | 239 (20.8) |
| Pulmonary stenosis | 80 (7.0) |
| Cyanotic congenital heart disease | |
| Tetralogy of Fallot | 53 (4.6) |
| Pulmonary atresia | 61 (5.3) |
| Double outlet right ventricle | 28 (2.4) |
| Transposition of the great arteries | 22 (1.9) |
| Tricuspid atresia | 11 (1.0) |
| Truncus arteriosus | 11 (1.0) |
| Mitral atresia | 7 (0.6) |
| Interrupted aortic arch | 3 (0.3) |
| Hypoplastic left heart syndrome | 1 (0.1) |

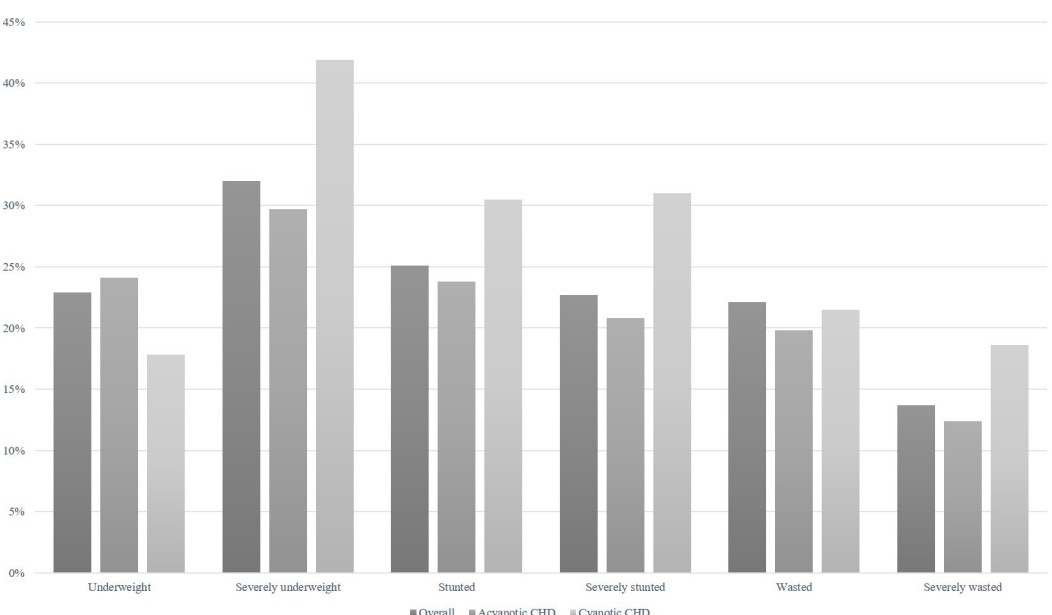

**Fig 1. Incidence of undernutrition in children with CHD.** Underweight, stunting and wasting were significantly associated with increased mortality with OR of 3.54 (95% CI: 1.62–7.74), p<0.001; 3.31 (1.65–6.64), *p*<0.001; and OR 1.64 (95% CI: 0.87–3.09), *p* = 0.129, respectively. In the multivariate analysis, delayed diagnosis of CHD, CHF, PH, presence of syndrome, low birth weight and being first child were independently associated with underweight (Table 2). The most common syndrome was Down syndrome.

The following variables were independently associated with stunting: cyanotic CHD, delayed diagnosis, CHF, PH, presence of syndrome, low maternal education, low birth weight and being first child (Table 3).

Further, factors that were independently associated with wasting included cyanotic CHD, delayed diagnosis of CHD, CHF, PH, and presence of syndrome (Table 4).

## Discussion

We observed a substantial burden of CHD in children in Yogyakarta, Indonesia, with over half of our cohort were categorized with undernutrition (underweight, stunting or wasting). As far

**Table 2. Factors associated with underweight.**

| Factors | Underweight n = 563 (%) | Normal n = 463 (%) | OR (95% CI) | *p* value | Adjusted OR (95%CI) |
|---|---|---|---|---|---|
| Male sex | 223 (39.6) | 252 (54.4) | 1.15 (0.90–1.47) | 0.285 | |
| Cyanotic CHD | 77 (13.7) | 114 (24.6) | 1.27 (0.93–1.75) | 0.147 | |
| Delayed diagnosis | 240 (42.6) | 338 (73) | 1.40 (1.09–1.79) | 0.009 | 1.69 (1.27–2.24) |
| CHF | 96 (17) | 201 (43.4) | 2.12 (1.60–2.82) | <0.001 | 1.76 (1.27–2.45) |
| Pulmonary hypertension | 27 (4.8) | 103 (22.2) | 3.62 (2.32–5.63) | <0.001 | 2.48 (1.50–4.10) |
| Syndrome | 27 (4.8) | 129 (27.8) | 4.8 (3.11–7.42) | <0.001 | 4.83 (3.07–7.62) |
| Low maternal education | 244 (43.3) | 281 (60.7) | 1.12 (0.87–1.43) | 0.380 | |
| Low birth weight | 102 (18.1) | 214 (46.2) | 2.17 (1.64–2.87) | <0.001 | 2.65 (1.95–2.61) |
| Being first child | 215 (38.2) | 192 (41.5) | 1.67 (1.30–2.16) | <0.001 | 1.61 (1.23–2.12) |

CHD, congenital heart disease; CHF, congestive heart failure; OR, odds ratio; CI, confidence interval.

**Table 3. Factors associated with stunting.**

| Factors | Stunting n = 549 (%) | Normal n = 600 (%) | OR (95%CI) | *p* value | Adjusted OR (95%CI) |
|---|---|---|---|---|---|
| Male sex | 273 (49.7) | 257 (42.8) | 0.95 (0.75–1.20) | 0.679 | |
| Cyanotic CHD | 82 (14.9) | 131 (21.8) | 1.98 (1.46–2.69) | <0.001 | 2.72 (1.93–3.83) |
| Delayed diagnosis | 256 (46.6) | 195 (32.5) | 1.35 (1.07–1.71) | 0.016 | 1.63 (1.20–2.22) |
| CHF | 254 (46.3) | 87 (14.5) | 1.33 (1.03–1.71) | 0.028 | 1.37 (1.00–1.86) |
| Pulmonary hypertension | 67 (12.2) | 99 (16.5) | 1.75(1.25–2.45) | 0.001 | 1.55 (1.03–2.33) |
| Syndrome | 34 (6.2) | 126 (21) | 4.96 (3.33–7.39) | <0.001 | 5.85 (3.84–8.92) |
| Low maternal education | 329 (59.9) | 262 (43.7) | 1.33(1.05–1.68) | 0.018 | 1.33 (1.04–1.72) |
| Low birth weight | 136 (24.7) | 195 (32.5) | 1.88 (1.45–2.44) | <0.001 | 2.91 (2.17–1.62) |
| Being first child | 278 (50.6) | 186 (31) | 1.69 (1.33–2.20) | <0.001 | 1.61 (1.24–2.08) |

CHD, congenital heart disease; CHF, congestive heart failure, CI, confidence interval; OR, odds ratio.

as we are aware, our study is among the first in Indonesia to evaluate the outcome and factors associated with undernutrition of children with CHD.

Our study showed that the proportion of undernutrition in children with CHD is high. This is similar to previous published studies [6–9]. It is known that risk factors for undernutrition among children with CHD include heart failure (HF), cyanosis, multiple heart defects, delayed corrective surgery, anemia, and pulmonary hypertension (PH). Feeding difficulties, increased basal metabolic rate, energy expenditure arising from cardiac defects, and hypoxia due to hemodynamic changes in these children contribute to undernutrition. In contrast, delayed corrective surgery for CHD increases the likelihood of the children developing undernutrition [6, 11, 12].

Children with severe HF had a higher risk of being stunting and wasting [7, 9]. Our studies found the same finding of HF as a risk factor for underweight, stunting, and wasting. Venous and bowel congestion due to hepatic or gastrointestinal dysfunction leads to early satiety and malabsorption. HF also activates the sympathetic nervous system leading to decreased appetite and increased metabolic demand. Restriction of fluid intake as a treatment for HF may have the unintended effect of excessive caloric restriction. Frequent respiratory infections and increased work of breathing alongside lower energy intake due to their inability to tolerate a high volume of feeding also play vital parts in the poor nutrition in these children [13, 14].

Incidence of undernutrition based on pathological type of CHD in our study was different from a previous study [9]. Remarkably, in our study the children with cyanotic CHD present

**Table 4. Factors associated with wasting.**

| Factors | Wasting n = 361 (%) | Normal n = 708 (%) | OR (95%CI) | *p* value | Adjusted OR (95%CI) |
|---|---|---|---|---|---|
| Male sex | 168 (46.5) | 338 (47.7) | 1.05 (0.81–1.35) | 0.746 | |
| Cyanotic CHD | 125 (34.6) | 84 (11.8) | 1.41 (1.04–1.93) | 0.034 | 1.56 (1.11–2.19) |
| Delayed diagnosis | 321 (88.9) | 274 (38.7) | 2.15 (1.62–2.85) | <0.001 | 1.80 (1.30–2.50) |
| CHF | 181 (50.1) | 145 (20.5) | 1.96 (1.49–2.56) | <0.001 | 1.70 (1.25–2.32) |
| Pulmonary hypertension | 81 (22.4) | 83 (11.7) | 2.31 (1.65–3.24) | <0.001 | 1.53 (1.04–2.25) |
| Syndrome | 87 (24.1) | 61 (8.6) | 1.45 (1.02–2.07) | 0.049 | 1.84 (1.26–2.70) |
| Low maternal education | 191 (52.9) | 366 (51.7) | 0.99 (0.66–1.49) | 0.746 | |
| Low birth weight | 157 (43.5) | 97 (13.7) | 1.29 (0.96–1.73) | 0.095 | |
| Being first child | 138 (38.2) | 301 (42.5) | 1.20 (0.92–1.55) | 0.189 | |

CHD, congenital heart disease; CHF, congestive heart failure, CI, confidence interval; OR, odds ratio.

as a risk factor in both episodes of wasting and stunting. This finding matches with previous studies that indicated stunting or wasting episodes are found in children with cyanotic CHD. However, some studies noted that only stunting was associated with cyanotic CHD, while others stated that only wasting was associated with cyanotic CHD [6, 7, 15, 16]. Both stunting and wasting are indicators of chronic malnutrition. Chronic hypoxia from right to left lesions with possible prolonged PH seen in cyanotic CHD induces direct and indirect effects on reduced serum hormone insulin-like growth factor I (IGF-I) that can cause impairment of bone center and eventually impair nutritional status and linear growth [17]. This could explain that cyanotic CHD was not a risk factor for underweight which marks acute undernutrition in our study.

In our study, both stunting and wasting were higher in cyanotic than in acyanotic CHD. These might because the proportion of delayed diagnosis among children with CHD were prominent, especially in those with cyanotic CHD. Therefore, children tend to present with complications related to delayed diagnosis of CHD including CHF or severe hypoxemia [10].

Undernutrition was significantly associated with increased risk of death in our study. Nutritional status may impact on the outcomes of cardiac surgery [18–20]. Children undergoing cardiac surgery, who are underweight, may be associated with a higher 30-day mortality. Among those who are stunting, it may be associated with longer duration of length of stay and mechanical ventilation as well as increased use of inotropes [21].

Cardiomyocytes can atrophy during starvation. Children with severe undernutrition may experience cardiomyopathy, HF, arrhythmia, and hypotension [2, 3, 22, 23]. Micronutrient deficiencies and electrolyte imbalances often occur under conditions of malnutrition and can affect cardiac function [2, 3, 22]. Severe iron deficiency anemia and thiamine deficiency can lead to high-output HF and vasodilation [23]. Selenium deficiency can reduce the function of cardiomyocytes. Severe hypophosphatemia can also reduce heart function and stroke volume. In addition, severe hypomagnesemia can increase the risk of arrhythmia [2].

Parents' height, single ventricle, and cyanotic heart disease were associated with stunting in a previous study [16]. A study from India revealed that the factors associated with stunting among children with CHD were small for gestation, lower maternal height and fat intake, and genetic syndromes [8]. Hospitalization and PH were associated with underweight and wasting [16]. A study from India revealed that the determinants of wasting among children with CHD were CHF, age at correction, lower birth weight and maternal weight, previous hospitalizations, religion and level of education of father [8].

Children with delayed diagnosis of CHD may develop complications including CHF and PH [10]. These eventually may lead to the development of underweight, stunting and wasting because of decreased energy intake, increased energy requirement, or both. CHF manifests as tachypnea and dyspnea may cause fatigue, decreases intake, anorexia, and malabsorption [24]. Some syndromes that are also present in children with CHD also impact the children's clinical stability, hence they become unfit for surgical intervention though CHD was already diagnosed [25]. Worse malnutrition also happens if the delayed surgical correction is for children with CHD under five years old [15, 26].

It is similar to our study that PH was associated with severe malnutrition in children with CHD. PH may cause undernutrition not only because of recurrent lower respiratory infections and CHF, but also chronic hypoxia and ineffective processing of nutrients in cellular level in cyanotic CHD [27]. Recent experimental studies suggest that besides reduced malabsorption of important nutrients, the gut microbiome is less diverse in adults with PH. Therefore, fewer metabolites are produced, impairing gut barrier function [28]. Gastrointestinal edema due to decreased right ventricular function will decrease nutrient uptake in patients with PH. Diuretics (such as furosemide) given to patients with PH will result in metabolic alkalosis and

hypokalemia that inhibit effective protein anabolism [6]. High doses of furosemide will also lead to thiamine deficiency, an essential vitamin for energy and carbohydrate regulation [29]. All these may explain PH as a risk factor in underweight, stunting, and wasting episodes [30].

Low birth weight is becoming a factor that independently associated with underweight and stunting among children with CHD in this study. Neonates with CHD are at greater risk for being born with low birth weight, premature, and small for gestational age [31, 32]. A meta-analysis suggests that various alterations in fetal hemodynamics and oxygen saturation due to CHD are the root causes of this association, such as reduced ventricular function that decreases cardiac output resulting in stunted fetal growth [33].

It was also interesting to note that mother's education level was only found significant in the stunting variables while in others, it was not. Our finding matched with several studies that showed generally, the lower the mother's level of education, the higher the chances of a mother having stunted children [34–36]. Education was an indirect factor in children's nutritional status because education level will affect the mother's parenting pattern. In a recent study of 70,293 Indonesian children, mothers in primary school and under education categories were more likely than mothers with a college education to have stunted children under two years [37].

Some of our findings were in line with previous references while some were not. Therefore, in this paper, we choose to mention each condition as a separate risk factor for undernutrition. These findings reveal considerable public health problems requiring comprehensive approach to improve outcome. The majority of factors associated with undernutrition found in this study are modifiable, and as such, may be considered in efforts to prevent the development of undernutrition among those with CHD to prevent infection, other morbidities, and improve outcomes [38]. These include early detection and screening of CHD, a reliable referral system, and comprehensive management including adequate nutritional intake, cardiac intervention and surgery, and cardiac intensive care program [39].

There are several limitations to our study. First, the nutritional status was assessed in various places such as outpatient clinic, PICU, NICU and pediatric wards. It may have influenced the current acute condition of patients related to nutritional status including medical equipment results in overestimation of weight measurements such as intravenous lines, pacemakers or other invasive devices. Second, our research was limited to Yogyakarta, Indonesia and as a single-center, our results cannot be generalized to other children with CHD in other low- and middle- income country settings.

## Conclusions

Our study demonstrates a considerable burden of undernutrition among children with CHD and this was significantly associated with increased risk of death. Factors associated with underweight, wasting and stunting are cyanotic CHD, delayed diagnosis, CHF, PH, presence of syndrome, history of low birth weight, and being first child. Accurate assessment of nutritional status is mandatory to obtain baseline and ongoing information on nutritional status in children with congenital heart disease.

## Supporting information

**S1 Dataset.**
(XLSX)

## Acknowledgments

The authors would like to Erik C Hookom for providing editorial assistance.

## Author Contributions

**Conceptualization:** Indah K. Murni, Nadya Arafuri, Sasmito Nugroho, Noormanto.

**Data curation:** Indah K. Murni, Linda Patmasari, M. Taufik Wirawan.

**Formal analysis:** Indah K. Murni, M. Taufik Wirawan.

**Investigation:** Indah K. Murni, Linda Patmasari, M. Taufik Wirawan.

**Methodology:** Indah K. Murni, M. Taufik Wirawan, Nadya Arafuri, Noormanto.

**Project administration:** Linda Patmasari, Esta Rossa Sativa.

**Resources:** Indah K. Murni, Neti Nurani.

**Supervision:** Indah K. Murni, Neti Nurani, Sasmito Nugroho, Noormanto.

**Validation:** Indah K. Murni, Linda Patmasari, M. Taufik Wirawan, Esta Rossa Sativa.

**Visualization:** Linda Patmasari.

**Writing – original draft:** Indah K. Murni, Linda Patmasari.

**Writing – review & editing:** Indah K. Murni, Linda Patmasari, M. Taufik Wirawan, Nadya Arafuri, Neti Nurani, Esta Rossa Sativa, Sasmito Nugroho, Noormanto.

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
