## [Decision Letter · Decision Letter 0]

16 Nov 2022

PONE-D-22-23981Outcome and factors associated with undernutrition among children with congenital heart diseasePLOS ONE

Dear Dr. Murni,

Thank you for submitting your manuscript to PLOS ONE. After careful consideration, we feel that it has merit but does not fully meet PLOS ONE’s publication criteria as it currently stands. Therefore, we invite you to submit a revised version of the manuscript that addresses the points raised during the review process.

Please revise. 

We look forward to receiving your revised manuscript.

Kind regards,

Academic Editor

PLOS ONE

Journal Requirements:

Reviewers' comments:

Reviewer's Responses to Questions

**Comments to the Author**

1. Is the manuscript technically sound, and do the data support the conclusions?

Reviewer #1: Yes

Reviewer #2: Partly

2. Has the statistical analysis been performed appropriately and rigorously? 

Reviewer #1: Yes

Reviewer #2: Yes

3. Have the authors made all data underlying the findings in their manuscript fully available?

Reviewer #1: Yes

Reviewer #2: Yes

4. Is the manuscript presented in an intelligible fashion and written in standard English?

Reviewer #1: Yes

Reviewer #2: No

5. Review Comments to the Author

Reviewer #1: The authors performed a cohort study to assess the nutritional status in children with CHD and to assess risk factors for this undernutrition. The topic is interesting and important. The methodology is sound and the manuscript is organized and well written but i have few comments:

1- the manuscript needs good English editing

2- what were the primary and secondary outcomes? Please add them at the end of the method section

3- what is the power of the study? And how did you estimate the sample size?

Reviewer #2: This study prospective investigated the burden of congenital heart disease (CHD) in Indonesia and nutritional status among children with CHD in one single center. The authors identified some risk factors associated with undernutrition. The case number in this study is large. I have some comments as below.

Major comments:

1. The definitions of underweight, stunting and wasting in methods were repetitive. Consider shorten the paragraph. The same problem occurred in discussion. The discussions of underweight, stunting and wasting in three separated paragraphs are redundant and repetitive.

2. The discussion is lengthy. Consider shorten the paragraph to half and avoid repetitive descriptions on each finding.

3. Some factors in table 2 should be defined, such as “syndrome” and “delayed diagnosis”. What diagnosis was included in syndrome or how to make a diagnosis of syndrome should be described.

4. Figure 1 is missing.

5. Why the risk factors of underweight, stunting and wasting are different? In-depth discussion is encouraged.

Minor comments:

1. “Multivariate” logistic regression is more suitable than “Multivariable” logistic regression.

2. Suggest replace “Syndrome” in abstract as a more specific term.

3. Abbreviations should be listed below each table.

4. Extensive English editing is needed.

6. PLOS authors have the option to publish the peer review history of their article (what does this mean?). If published, this will include your full peer review and any attached files.

Reviewer #1: No

Reviewer #2: No

---

## [Author Response · Author response to Decision Letter 0]

2 Jan 2023

Reviewer #1: The authors performed a cohort study to assess the nutritional status in children with CHD and to assess risk factors for this undernutrition. The topic is interesting and important. The methodology is sound and the manuscript is organized and well written but i have few comments:

1- the manuscript needs good English editing

Response to Reviewer's comment:

We thank you very much for the positive comments and interest in the paper.

We have consulted the manuscript to a native English speaker:

Erik Christopher Hookom, BA, M.Ed, TEFL.

Office of Research and Publication (ORP)

Faculty of Medicine, Public Health and Nursing, Universitas Gadjah Mada

Administration Building 2nd Floor

Phone: +62 274 560300 ext 205

Email: echookom@gmail.com

We have carefully revised the manuscript to address the errors and grammatical mistakes throughout the paper. 

2- what were the primary and secondary outcomes? Please add them at the end of the method section

Responses to Reviewers:

Thank you very much for your suggestion. We have included the primary and secondary outcome at the end of the method section. We added “The primary outcome of this study was the outcome of undernutrition among children with CHD. The secondary outcome were factors associated with undernutrition among children with CHD.”

3- what is the power of the study? And how did you estimate the sample size?

Responses to Reviewers:

We have tried to calculate the estimated sample size using this formula below with 80% of power and the number of subjects included in our study exceeded the estimated sample size. 

The prevalence of malnutrition in children with congenital heart disease based on a previous study by Batte et al. 2017 is 31.5% - 45.4%. 

Sample size= (z^2 x p (1-p))/c^2 

z=z value z=z value (1.96 for 95% confidence level)

p=proportion of malnutrition in children with CHD (estimated 31%)

c=margin of errors (expressed in decimal)(0.05)

Sample size= (〖1.96〗^2 x 0.31 (1-0.31))/〖0.05〗^2 

Sample size=329

Reviewer #2: This study prospective investigated the burden of congenital heart disease (CHD) in Indonesia and nutritional status among children with CHD in one single center. The authors identified some risk factors associated with undernutrition. The case number in this study is large. I have some comments as below.

Major comments:

1. The definitions of underweight, stunting and wasting in methods were repetitive. Consider shorten the paragraph. The same problem occurred in discussion. The discussions of underweight, stunting and wasting in three separated paragraphs are redundant and repetitive.

Responses to Reviewers:

We thank you very much for the positive comments, suggestion, and interest in the paper.

We have shortened the paragraph into “The nutritional status was determined using criteria which were derived from the standards of the World Health Organization (WHO). Undernutrition included underweight, stunting and wasting. “

2. The discussion is lengthy. Consider shorten the paragraph to half and avoid repetitive descriptions on each finding.

Thank you very much for your suggestion. We have revised and shorten the discussion section.

3. Some factors in table 2 should be defined, such as “syndrome” and “delayed diagnosis”. What diagnosis was included in syndrome or how to make a diagnosis of syndrome should be described.

Thank you very much for your suggestion. We have included this sentence into the Methods: 

“Presence of genetic syndrome was defined when patients were diagnosed as having any congenital dysmorphic syndrome. The diagnosis of syndrome was defined based on clinical appearance of the patients”. The most common syndrome was Down syndrome. 

“Delayed diagnosis was defined based on their pathologic type: cyanotic CHD and acyanotic CHD. Delayed diagnosis in acyanotic CHD was defined at an age where elective cardiac repair should have already been performed or in case immediate treatment was indicated because of the patient hemodynamic status. Patient with delayed diagnosis in cyanotic CHD were defined as newborns discharged from their birth clinic or hospital without a CHD diagnosis”.

4. Figure 1 is missing.

Thank you. We have submitted the figure 1 in different file.

5. Why the risk factors of underweight, stunting and wasting are different? In-depth discussion is encouraged.

Thank you very much for raising this. The presence of undernutrition was associated with increased risk of death among children with CHD. Some of our findings were in line with previous references while some were not. For example, it is interesting to note in our study that children with cyanotic CHD present as a risk factor in both episodes of wasting and stunting. This is in line with studies from Chinawa et al. in Nigeria, Basheir et al. in Egypt, Zhang et al. in China, and Okoromah et al. in Nigeria that stunting or wasting episodes are found in children with cyanotic CHD. However, some studies noted that only stunting was associated with cyanotic CHD, while others stated that only wasting was associated with cyanotic CHD (6,7,15,16). Both stunting and wasting are indicators of chronic malnutrition. Chronic hypoxia from right to left lesion with possible prolonged pulmonary hypertension seen in cyanotic CHD provides direct and indirect effects on reduced serum hormone insulin-like growth factor I (IGF-I) that can cause impairment of bone center and eventually impair nutritional status and linear growth (17). This could explain why in our study, cyanotic CHD was not a risk factor for underweight which marks acute undernutrition. Therefore, in this paper, we choose to mention each condition as a separate risk factor.

Minor comments:

1. “Multivariate” logistic regression is more suitable than “Multivariable” logistic regression.

Thank you very much for your suggestion. We have revised it in abstract and methods section.

2. Suggest replace “Syndrome” in abstract as a more specific term.

Thank you very much for your suggestion. We have revised as suggested.

3. Abbreviations should be listed below each table.

Thank you very much for your suggestion. We have listed the abbreviation below each table in the manuscript. 

4. Extensive English editing is needed.

Thank you very much for your suggestion. We have consulted the manuscript to a native English speaker:

Erik Christopher Hookom, BA, M.Ed, TEFL.

Office of Research and Publication (ORP)

Faculty of Medicine, Public Health and Nursing, Universitas Gadjah Mada

Administration Building 2nd Floor

Phone: +62 274 560300 ext 205

Email: echookom@gmail.com

We have carefully revised the manuscript to address the errors and grammatical mistakes throughout the paper.

---

## [Decision Letter · Decision Letter 1]

1 Feb 2023

Outcome and factors associated with undernutrition among children with congenital heart disease

PONE-D-22-23981R1

Dear Dr. Murni,

We’re pleased to inform you that your manuscript has been judged scientifically suitable for publication and will be formally accepted for publication once it meets all outstanding technical requirements.

Kind regards,

Academic Editor

PLOS ONE

Additional Editor Comments (optional):

Reviewers' comments:

Reviewer's Responses to Questions

**Comments to the Author**

1. If the authors have adequately addressed your comments raised in a previous round of review and you feel that this manuscript is now acceptable for publication, you may indicate that here to bypass the “Comments to the Author” section, enter your conflict of interest statement in the “Confidential to Editor” section, and submit your "Accept" recommendation.

Reviewer #1: All comments have been addressed

Reviewer #2: All comments have been addressed

2. Is the manuscript technically sound, and do the data support the conclusions?

Reviewer #1: Yes

Reviewer #2: Yes

3. Has the statistical analysis been performed appropriately and rigorously? 

Reviewer #1: Yes

Reviewer #2: Yes

4. Have the authors made all data underlying the findings in their manuscript fully available?

Reviewer #1: Yes

Reviewer #2: Yes

5. Is the manuscript presented in an intelligible fashion and written in standard English?

Reviewer #1: Yes

Reviewer #2: Yes

6. Review Comments to the Author

Reviewer #1: the authors performed the required changes and the manuscript is now ready to be accepted and published.

Reviewer #2: Thank you for inviting me to reviewer this resubmitted manuscript. The authors have addressed all the issues I mentioned. I have no further comment on this manuscript.

7. PLOS authors have the option to publish the peer review history of their article (what does this mean?). If published, this will include your full peer review and any attached files.

Reviewer #1: **Yes: **Doaa El Amrousy

Reviewer #2: No

---

## [Editor Report · Acceptance letter]

14 Feb 2023

PONE-D-22-23981R1 

Outcome and factors associated with undernutrition among children with congenital heart disease 

Dear Dr. Murni:

I'm pleased to inform you that your manuscript has been deemed suitable for publication in PLOS ONE. Congratulations! Your manuscript is now with our production department. 

Kind regards, 

on behalf of

Dr. Robert Jeenchen Chen 

Academic Editor

PLOS ONE